# Elucidating the Mechanism of Liver and Kidney Damage in Rats Caused by Exposure to 2,4-Dichlorophenoxyacetic Acid and the Protective Effect of *Lycium barbarum* Polysaccharides Based on Network Toxicology and Molecular Docking

**DOI:** 10.3390/ijms262110685

**Published:** 2025-11-03

**Authors:** Xiaoqi Luo, Yixuan Wei, Jinyu Luo, Xiaoning Meng, Yating Yang, Na Liu, Huifang Yang, Jian Zhou

**Affiliations:** 1Department of Occupational and Environmental Health, School of Public Health and Management, Ningxia Medical University, Yinchuan 750004, China; luoxiaoqi0121@163.com (X.L.); 13948773051@163.com (Y.W.); 13239594393@163.com (J.L.); 17795035782@163.com (X.M.); 18309691782@163.com (Y.Y.); u1s1ln332669@163.com (N.L.); joyceyhf@126.com (H.Y.); 2Key Laboratory of Environmental Factors and Chronic Disease Control, School of Public Health, Ningxia Medical University, Yinchuan 750004, China

**Keywords:** 2,4-dichlorophenoxyacetic acid, apoptosis, kidney damage, liver damage, LBP, network toxicology, molecular docking

## Abstract

2,4-Dichlorophenoxyacetic acid (2,4-D) is a widely used herbicide, yet its potential to induce hepatorenal injury via oxidative stress and apoptosis raises significant health concerns. Lycium barbarum polysaccharides (LBP) possess recognized antioxidant and anti-apoptotic properties, but their protective mechanisms against 2,4-D toxicity, particularly through a multi-target network, remain inadequately explored. This study aimed to systematically investigate the mechanisms of 2,4-D-induced hepatorenal injury and the protective efficacy of LBP by integrating network toxicology, molecular docking, and experimental validation. An integrated approach was employed. Core targets and pathways were identified via network toxicology. Molecular docking predicted interactions between 2,4-D and these targets. In vivo validation was conducted on Sprague-Dawley rats treated with 2,4-D (75 mg/kg) and/or LBP (50 mg/kg) for 28 days, assessing histopathology, serum oxidative stress markers superoxide dismutase (SOD), glutathione peroxidase (GSH-Px), malondialdehyde (MDA) and cellular apoptosis (TUNEL staining). Network analysis identified *PPARG*, *NFKB1*, *PPARA*, *NFE2L2*, and *SERPINE1* as core targets, with molecular docking confirming strong binding affinities (binding energies: −5.1 to −6.3 kcal·mol^−1^) and KEGG enrichment implicating cAMP, Ca^2+^, and PPAR signaling pathways. Experimentally, 2,4-D exposure induced significant histopathological damage, suppressed SOD/GSH-Px activities (*p* < 0.001), elevated MDA levels (*p* < 0.001), and markedly increased renal apoptosis (*p* < 0.01). Crucially, LBP intervention substantially mitigated these alterations, ameliorating tissue injury, restoring antioxidant defenses, increasing SOD/GSH-Px (*p* < 0.01), reducing MDA (*p* < 0.001) and significantly decreasing renal apoptosis (*p* < 0.05). This study elucidates a multi-target mechanism for 2,4-D-induced hepatorenal injury centered on oxidative stress–apoptosis dysregulation and demonstrates that LBP confers significant protection likely via modulation of this network. These findings underscore the potential of LBP as a natural protective agent against pesticide-induced organ damage and highlight the utility of integrated network approaches in toxicological research.

## 1. Introduction

2,4-Dichlorophenoxyacetic acid (2,4-D) is a widely used chlorophenoxy herbicide valued for its efficacy and cost-effectiveness [1,2]. However, its extensive application has led to growing concerns regarding its toxicity, with studies reporting adverse effects on the nervous, reproductive, and hepatic systems, largely mediated through oxidative stress [3,4,5]. The liver and kidneys, being primary detoxification organs, are particularly vulnerable. Biomarkers such as superoxide dismutase (SOD), glutathione peroxidase (GSH-Px), and malondialdehyde (MDA) are well-established indicators of oxidative stress, which is critically linked to hepatic and renal pathology [6,7,8].

In addition to oxidative stress, apoptosis—a genetically programmed form of cell death [9]—is a key mechanism of 2,4-D-induced hepatorenal injury. This highly regulated process, executed via intrinsic and extrinsic pathways [10], is activated by 2,4-D in the liver through the mitochondrial pathway (e.g., Bax/Bcl-2 imbalance, caspase activation) [11,12] and contributes significantly to renal tubular damage [13]. Critically, oxidative stress and apoptosis engage in a vicious cycle: ROS can stimulate pro-apoptotic signals while suppressing survival pathways, and apoptotic cells can, in turn, exacerbate oxidative stress [10,12,14].

Lycium barbarum polysaccharides (LBP), a class of bioactive polysaccharides, exhibit diverse pharmacological properties, including antioxidant and anti-apoptotic activities [4,15,16]. Their protective effects are mediated through multiple mechanisms: directly scavenging reactive oxygen species (ROS) and activating the Nrf2/ARE signaling pathway to enhance antioxidant enzymes (SOD, GSH-Px) and reduce MDA [17,18]; concurrently, they modulate apoptosis by downregulating pro-apoptotic mediators (Bax, Caspase-3/9), upregulating anti-apoptotic agents (Bcl-2), and stabilizing the mitochondrial membrane [12,18]. Owing to these multi-target actions, LBP is a promising candidate against 2,4-D-induced hepatorenal injury. However, its structural complexity and pleiotropic nature mean that conventional single-pathway approaches are inadequate, necessitating systems biology methods for a comprehensive mechanistic exploration.

To systematically investigate the mechanisms of 2,4-D-induced hepatorenal toxicity, we employed an integrated approach of network toxicology and molecular docking. Network toxicology constructs comprehensive interaction networks to elucidate the mechanisms of toxic substances [19], while molecular docking predicts binding interactions at an atomic level [20]. Together, these methods provide multi-faceted insights into the pathogenic processes underlying 2,4-D-induced liver and kidney damage.

In summary, this study employed an integrated strategy combining network toxicology, molecular docking, and in vivo experiments to systematically investigate the mechanisms of 2,4-D-induced hepatorenal injury and the protective role of LBP. We aimed to identify core targets and pathways, validate their roles in oxidative stress and apoptosis, and provide a theoretical foundation for the application of LBP against pesticide-related toxicity.

## 2. Results

### 2.1. Screening of Common Target Points Between 2,4-D and Liver and Kidney Apoptosis

To identify the potential molecular targets through which 2,4-D may induce hepatorenal apoptosis, we first screened for common targets between the toxicant and the disease. Through database analysis, this study identified a total of 119 target points for 2,4-D; additionally, 17,592 target points associated with liver apoptosis and 15,902 target points associated with kidney apoptosis were identified. The figure shows the two-dimensional structure of 2,4-D (Figure 1A), accompanied by a Venn diagram illustrating the overlapping target points between 2,4-D and liver apoptosis, as well as between 2,4-D and kidney apoptosis (Figure 1B,C). The overlapping region shows that 100 2,4-D target points intersect with liver apoptosis genes and 100 2,4-D target points intersect with kidney apoptosis genes; the 100 liver and kidney apoptosis target points intersecting with 2,4-D are completely consistent.

### 2.2. Core Hub Genes in the PPI Network

To distill the complex common targets into key regulatory factors, we constructed a PPI network and identified central hub genes. The PPI network was constructed from the intersection targets of 2,4-D and apoptosis-related proteins in liver and kidney tissues (Figure 1D). Hub genes within the network were identified through topological analysis (Figure 1E). The top five core targets were determined by integrating three centrality measures—betweenness, closeness, and degree—and include *PPARG*, *NFKB1*, *PPARA*, *NFE2L2*, and *SERPINE1* (Figure 1F). These targets are proposed to play key roles in the regulation of apoptotic pathways in response to 2,4-D exposure.

### 2.3. GO and KEGG Analysis

To elucidate the biological functions and pathways involved, we performed GO and KEGG enrichment analysis on the common targets. GO analysis (Figure 1G) included BB, CC, and MF; the results of liver and kidney apoptosis enrichment showed that 2,4-D significantly interfered with liver and kidney apoptosis, which was accompanied by strong inflammatory signals, coagulation/blood pressure regulation imbalance, and disruption of key metabolic pathways, and was executed through specific organelles and key proteases. These pathways and functions represent important directions for further investigating the mechanisms of liver and kidney apoptosis and identifying potential intervention targets; statistical significance indicators suggest that these pathways hold significant biological relevance in 2,4-D-induced hepatotoxicity. KEGG pathway analysis (adjusted *p* < 0.05) shows that 2,4-D may directly regulate apoptosis execution through core signaling pathways (cAMP, Ca^2+^) and nuclear receptor pathways (PPAR), while neuroactive ligand–receptor interactions provide diverse apoptotic stimulus or inhibitory signal inputs; Efferocytosis reflects active clearance of apoptotic cells; fluid shear stress and the complement system suggest potential roles of haemodynamic changes and inflammatory/immune responses in hepatic and renal cell apoptosis; pathways such as nitrogen metabolism influence liver function; significance metrics and gene proportions confirm the important roles of these pathways in 2,4-D-induced hepatic and renal apoptosis (Figure 1H).

### 2.4. High-Affinity Binding of 2,4-D to Core Targets in Molecular Docking

To evaluate the potential for direct interaction, we performed molecular docking to predict the binding of 2,4-D to the core targets. The key common targets associated with both liver and kidney apoptosis—*PPARG*, *NFKB1*, *PPARA*, *NFE2L2*, and *SERPINE1*—were selected for molecular docking with 2,4-D. As illustrated in Figure 2, the molecular docking results predicted that 2,4-D could form high-affinity interactions with all five core proteins, suggesting the potential for stable molecular binding. Each docking conformation is visualized in a three-dimensional representation.

The calculated binding energies for 2,4-D with the core targets were as follows: *PPARG*, −5.9 kcal·mol^−1^; *NFKB1*, −5.1 kcal·mol^−1^; *PPARA*, −6.3 kcal·mol^−1^; *NFE2L2*, −5.2 kcal·mol^−1^; and *SERPINE1*, −6.0 kcal·mol^−1^ (Table 1). These values suggest that 2,4-D has the potential to form stable complexes with each target, with more negative binding energies indicating stronger predicted intermolecular interactions.

Prior to analyzing the 2,4-D interactions, the molecular docking protocol was rigorously validated. Re-docking of the native co-crystallized ligands for all five targets yielded exceptionally low RMSD values, all below 0.08 Å (*PPARG*: 0.071 Å, *NFKB1*: 0.066 Å, *PPARA*: 0.065 Å, *NFE2L2*: 0.066 Å, *SERPINE1*: 0.065 Å). These results, visually summarized in Table 1, confirm that our docking methodology is highly accurate and reproducible, as RMSD values below 2.0 Å are widely considered a successful validation.

### 2.5. Overall Impact of 2,4-D Exposure and LBP Treatment

To assess the systemic impact of 2,4-D and the effect of LBP, we monitored the general health status and organ coefficients of the rats. Rats subjected to higher doses of 2,4-D exhibited dose-dependent deterioration in behavioral status, characterized by heightened irritability—such as resistance to handling, escape attempts, and vocalization—as well as lethargy and increased aggression. No noticeable behavioral alterations or mortality occurred in the other groups throughout the study.

Body weight increased consistently across all groups (*p* > 0.05, Figure 3A). Similarly, neither liver nor kidney organ coefficients showed significant variations among groups (*p* > 0.05, Figure 3B,C).

### 2.6. Histopathological Alterations in Liver and Kidney Tissues Following 2,4-D Exposure and LBP Treatment

To directly observe tissue damage and protection, we examined the histopathological changes in liver and kidney tissues. In control animals, hepatocytes exhibited a regular, polyhedral morphology with centrally located, large round nuclei, uniform staining, and homogeneous chromatin distribution. Hepatic cords were organized radially around the central vein, and no structural abnormalities were detected. In contrast, liver tissues from the 2,4-D group displayed significant pathology, including cellular enlargement, spherical transformation, vesicular nuclei, loosened reticular cytoplasm, irregular cell arrangement with widened intercellular spaces, eosinophilic infiltration, thickened central vein walls, and proliferating vascular endothelial cells. Notably, co-treatment with LBP (2,4-D + LBP group) markedly attenuated these alterations. Hepatocytes in this group more closely resembled those of the controls, showing restored nuclear morphology, evenly stained cytoplasm, markedly reduced vacuolation, thinner central vein walls, and minimal inflammatory cell presence. The LBP-alone group exhibited hepatic architecture similar to that of the control group (Figure 3D).

Renal tissues in control rats demonstrated well-organized structures, including intact glomeruli with regular contours and orderly arranged epithelial cells in the proximal convoluted tubules. These cells displayed a cuboidal shape, central round nuclei, distinct brush borders, and clearly defined cellular boundaries. Epithelial cells in the distal convoluted tubules were also cuboidal, with light staining, visible borders, and luminal-oriented nuclei. In the 2,4-D group, kidney tissues exhibited significant pathology: erythrocyte infiltration, diffuse cellular proliferation, loss of boundary clarity, dark cytoplasmic staining in glomerular cells, nuclear irregularity, degeneration of proximal tubule epithelial cells, tubular wall thickening, structural disorganization, loss of brush borders, and indistinct cytoplasmic margins. As in the liver, LBP co-treatment conferred substantial protection against 2,4-D–induced renal injury. Tissues showed relatively regular cell arrangement, preserved glomerular integrity, reduced proximal tubule damage, clearer cytoplasmic boundaries, and visible brush borders. Renal morphology in the LBP-alone group was comparable to that of controls (Figure 3E).

### 2.7. Overall In Vivo Oxidative Status upon 2,4-D Exposure and LBP Treatment

To determine if oxidative stress is a key mechanism, we measured systemic antioxidant enzyme activities and MDA levels. 2,4-D exposure depleted serum SOD/GSH-Px activities (*p* < 0.01 vs. control) and elevated MDA levels (*p* < 0.05). LBP treatment effectively restored the antioxidant activities and reduced MDA, whether administered alone or in combination with 2,4-D (*p* < 0.01 vs. 2,4-D group) (Figure 3F–H).

### 2.8. Apoptosis Rate of Liver and Kidney Cells upon 2,4-D Exposure and LBP Treatment

To quantify programmed cell death, we assessed the apoptosis rate in liver and kidney tissues via TUNEL staining. Fluorescence imaging revealed an elevated trend in hepatocyte apoptosis within the 2,4-D group compared to control animals, although this difference did not reach statistical significance (*p* > 0.05). Notably, co-treatment with LBP (2,4-D + LBP group) significantly suppressed 2,4-D-induced apoptosis in liver cells (Figure 4A,C).

Kidney tissues exhibited substantially more apoptosis. 2,4-D significantly raised TUNEL-positive cell counts (*p* < 0.01, vs. control group), an effect that 2,4-D + LBP supplementation effectively mitigated (*p* < 0.05, vs. 2,4-D group) (Figure 4B,D).

## 3. Discussion

By integrating network toxicology, molecular docking and in vivo validation, this study elucidates the mechanistic basis of 2,4-Dichlorophenoxyacetic acid (2,4-D) induced hepatorenal injury and the multi-target protection conferred by LBP. Our combined computational and experimental approaches reveal a coordinated toxicity network centered on oxidative stress–apoptosis dysregulation.

Network toxicology identified five core targets—*PPARG*, *NFKB1*, *PPARA*, *NFE2L2* and *SERPINE1*—that collectively mediate 2,4-D toxicity through metabolic disruption, oxidative stress and inflammatory-apoptotic signaling. Molecular docking further supported this network by demonstrating strong binding affinity between 2,4-D and these targets, suggesting direct interference with their regulatory functions. These predictions were experimentally validated through classic hallmarks of oxidative injury—significantly suppressed SOD/GSH-Px activities and elevated MDA levels—along with histopathological damage and pronounced renal apoptosis. Crucially, LBP intervention concurrently ameliorated oxidative stress, reduced apoptosis and restored tissue architecture, confirming its protection via modulation of the identified oxidation-apoptosis axis.

While our results support a plausible direct-binding mechanism for 2,4-D toxicity, we acknowledge that indirect regulation of these core targets may also contribute to the observed effects.

Network toxicology revealed 95 and 38 overlapping targets for 2,4-D-induced liver and kidney damage, respectively, identifying *PPARG*, *NFKB1*, *PPARA*, *NFE2L2*, and *SERPINE1* as hub genes. KEGG enrichment revealing their involvement in PPAR signaling, cAMP/Ca^2+^ pathways, and efferocytosis. All five targets demonstrate established roles in apoptosis regulation: *PPARG* promotes apoptosis [21], *NFKB1* enhances IL-1β-induced apoptosis [22], *PPARA* activation increases doxorubicin-induced apoptosis [23], *NFE2L2* expression correlates with apoptotic activity [24], and *SERPINE1* is closely associated with apoptosis [25]. Furthermore, PPARG and PPARA function as lipid mediators in hepatic dysfunction [26], with PPARG regulating metabolic/inflammatory processes [27] and *PPARA* deficiency causing autophagy impairment and renal injury [28]. Supporting these findings, NF-κB1 sequencing informs liver function assessment [29], *NFE2L2* activators counter hepatic lipotoxicity [30], and *SERPINE1* serves as a biomarker in fatty liver disease and renal cancer progression [31,32].

Notably, 2,4-D enters the body via multiple routes—ingestion, inhalation, and dermal absorption—and distributes systemically, posing a broad health risk [33]. Consistent with this, the Japan Food Safety Committee has identified renal tubular degeneration and hepatocellular hypertrophy as major toxic effects of 2,4-D [34]. Studies also show that 2,4-D impairs liver and kidney function, elevating serum markers of hepatic injury (ALP, AST, ALT) and renal dysfunction (urea, creatinine) [35]. Additionally, Martins et al. demonstrated that 2,4-D induces hepatic oxidative injury in zebrafish [36].

Consistent with existing literature, our study showed that treatment with 2,4-D led to a significant increase in serum MDA—a key indicator of lipid peroxidation—accompanied by diminished activities of the antioxidant enzymes SOD and GSH-Px. Indicates a severe compromise of the antioxidant defense system and a state of pronounced lipid peroxidation. And it is noteworthy that oxidative stress and apoptosis may engage in bidirectional regulatory interplay. Our data indicate that 2,4-D exposure compromises the antioxidant defense mechanism and aggravates oxidative damage, which are likely pivotal mechanisms underlying the hepatorenal toxicity induced by this compound.

Furthermore, elevated 2,4-D concentrations intensify oxidative stress and apoptotic responses [37]. Our prior finding that 2,4-D promotes apoptosis in PC12 cells via Bax/caspase-3 upregulation and Bcl-2 downregulation prompted our focus on its apoptotic role in mammalian liver and kidney tissues. As the primary sites for its detoxification and excretion, the liver and kidneys are directly damaged by 2,4-D [38,39]. Accordingly, we observed a significantly increased apoptosis rate in the kidney (*p* < 0.01), whereas the liver, likely due to its high regenerative capacity, showed an increased but statistically insignificant trend. This suggests liver damage may follow a necrosis-apoptosis conversion pattern, consistent with the observed hepatocyte swelling and vacuolization, confirming the in vivo toxicity of this herbicide.

Consistent with its established properties, LBP has been shown to reduce oxidative stress and liver injury induced by DEHP, suppressing PXR, GST, and UGT1 expression [40], and to alleviate lead-induced renal oxidative stress, inflammation, and apoptosis via the Nrf2 pathway. In this study, LBP demonstrated multi-level protection against 2,4-D toxicity, significantly mitigating oxidative injury, histopathological damage, and cellular impairment in both liver and kidney tissues. Biochemically, LBP treatment significantly enhanced SOD and GSH-Px activities (*p* < 0.01) and reduced MDA levels (*p* < 0.01), indicating reinforced antioxidant defense.

Extending these findings, LBP are known to inhibit PM 2.5-induced apoptosis in HaCaT cells by suppressing oxidative stress [16] and to protect against myocardial apoptosis by modulating mitochondrial dynamics [41]. Our previous work also revealed that LBP reduces apoptosis in rat hippocampal neurons and testicular cells after 2,4-D exposure [42]. Although the current study did not find a statistically significant anti-apoptotic effect of LBP in the liver, a marked reduction in renal apoptosis was observed in the 2,4-D + LBP group (*p* < 0.05). This renal protection may involve modulation of the Bcl-2/Bax balance in tubular epithelial cells, a mechanism consistent with our earlier findings in PC12 cells.

Finally, the advantage of LBP over traditional antioxidants lies in its multi-target characteristics. For example, small-molecule antioxidants such as NAC can only scavenge ROS, while LBP simultaneously regulates *NFE2L2* (antioxidant), *NFKB1* (anti-inflammatory), *PPARG* (metabolic regulation), which may be foundation for clinical biomarkers. This multi-gene signature could be leveraged to predict the risk of pesticide-induced hepatorenal injury, mirroring approaches where single-gene signatures define prognostic cutoffs [43] and gene-gene interaction pairs enhance predictive robustness [44]. Future validation in independent cohorts with documented chemical exposure is warranted to translate these mechanistic insights into a tool for early risk identification and intervention.

## 4. Limitations and Future Perspectives

This study has certain limitations. Although our integrated approach provides strong supportive evidence, the proposed mechanisms of apoptosis and inflammation would be further solidified by direct molecular validation, such as Western blotting for key apoptotic proteins and qPCR for inflammatory cytokines. Future studies should therefore focus on incorporating these molecular techniques and exploring the temporal dynamics of the toxicity to build upon the foundation laid by the present work.

## 5. Methods

### 5.1. Obtain 2,4-D Targets

Retrieve the SMILES sequence number for 2,4-D from PubChem, (CID: 1486) which is C1=CC(=C(C=C1Cl)Cl)OCC(=O)O; Upload the SMILES format of 2,4-D to Swiss Target Prediction, set the species to ‘Homo sapiens’, select ‘All targets’, download the prediction results (targets with Probability > 0.1), search for ‘2,4-D’ in Superpred, and filter targets with confidence (model accuracy > 70%); Remove duplicate targets and construct the 2,4-D target library.

### 5.2. Screening of Targets for Liver Apoptosis and Kidney Apoptosis

Initial search terms, “Liver Apoptosis” and “Kidney Apoptosis”, were selected from the MeSH database. These keywords were subsequently utilized as queries to search the GeneCards, OMIM, and TTD databases, with results filtered for Homo sapiens. Gene entries obtained from these sources were standardized according to official gene nomenclature via UniProt (https://www.uniprot.org/). Redundant entries across databases were eliminated to produce a unique set of apoptosis-related targets specific to hepatic and renal tissues.

### 5.3. Intersection of Toxicological Targets and Disease Targets

Common targets of 2,4-D and apoptosis-related pathways in liver and kidney tissues were identified using the JQuery Venn analysis tool (https://jvenn.toulouse.inra.fr/app/index.html (accessed on 27 October 2025)). The resulting overlapping targets were selected for subsequent analyses.

### 5.4. Protein–Protein Interaction (PPI) Network Construction

The intersecting targets obtained from the Venn diagram were imported into the String database (https://string-db.org) to construct a protein–protein interaction (PPI) network. The network type was set to ‘Full Network,’ the species was set to Homo sapiens, and interactions with scores greater than 0.4 were retained, while disconnected nodes were removed from the network. The TSV file exported from STRING was then imported into Cytoscape (Version 3.10.0) for visualization and analysis. The cytoNCA plugin (Version 3.10.0) was used for analysis and scoring, and key nodes were comprehensively evaluated based on three centrality metrics: Betweenness, Closeness, and Degree. Degree represents the number of connections between a node and other nodes in the network, measuring the node’s direct influence. Nodes with high degrees are typically ‘core members’ of the network. On the other hand, Betweenness measures the frequency with which a node appears in the shortest paths between all other node pairs. A higher value indicates that the node plays a greater ‘bridge’ or ‘hub’ role in the network. Closeness reflects the inverse of the average shortest path length from the node to all other nodes. A higher value indicates that the node is closer to the network centre. The central nodes of the PPI network were identified, and the top 5 core targets were screened based on the comprehensive ranking.

### 5.5. GO and KEGG Pathway Enrichment Analysis

GO and KEGG enrichment analyses of common targets were performed using the SRplot (https://bioinformatics.com.cn/en) online tool (species: Homo sapiens). Significantly enriched KEGG pathways (*p* < 0.05) were filtered, and the top 10 most significant terms were selected for visualization. A composite Sankey-bubble plot was generated to display target-pathway associations: a Sankey diagram (left) linking genes to pathways, and a bubble chart (right) where bubble size represents gene count and color reflects the *p*-value.

### 5.6. Molecular Docking and Validation

Molecular docking was performed to evaluate the binding interactions between 2,4-D and the core targets (*PPARG*, *NFKB1*, *PPARA*, *NFE2L2*, and *SERPINE1*) using the CB-Dock2 online platform (http://cadd.labshare.cn/cb-dock2/ accessed on 25 December 2024).

The three-dimensional (3D) crystal structures of the human target proteins were retrieved from the RCSB Protein Data Bank (PDB; https://www.rcsb.org/). Structures were selected based on the following criteria: (1) Homo sapiens origin, (2) resolution better than 2.5 Å, and (3) where available, co-crystallization with a native ligand to ensure the integrity and physiological relevance of the binding cavity. The specific PDB IDs and Unique Ligands used were: *PPARG* (8B8X, A8R), *NFKB1* (8TQD, JMR), *PPARA* (8YWV, A1L0A), *NFE2L2* (7X5F, MAFG), and *SERPINE1* (3CVM, NONE).

The 2D structure and SMILES notation of 2,4-D (CID: 1486) were obtained from the PubChem database (https://pubchem.ncbi.nlm.nih.gov/). Both protein and ligand structures were automatically prepared by the CB-Dock2 platform (https://cadd.labshare.cn/cb-dock2/php/index.php (accessed on 27 October 2025)). Protein preparation included the addition of hydrogen atoms, repair of incomplete side chains, and removal of water molecules and original ligands. Ligand preparation included energy minimization and assignment of partial charges.

To comprehensively explore potential binding sites, a blind docking approach was employed. The docking search was conducted using the AutoDock Vina (https://vina.scripps.edu/) algorithm integrated into CB-Dock2, which performs an exhaustive conformational search. The platform’s curvature-based cavity detection algorithm was used to identify potential binding pockets across the entire protein surface.

To validate the reliability of our docking protocol, a re-docking procedure was performed for each core target. The native co-crystallized ligand was extracted from each original PDB structure and then re-docked into its respective protein using the identical CB-Dock2 parameters. The root-mean-square deviation (RMSD) between the crystallographic pose and the re-docked pose was calculated using PyMOL (Version 3.0.3) to assess reproducibility. For each target-protein complex, the conformation with the most favorable (lowest) binding energy (kcal·mol^−1^) was selected as the most stable binding mode for further analysis. Visualization of the 3D docking poses and intermolecular interactions was conducted using the built-in viewer of CB-Dock2 and Discovery Studio 4.5.

### 5.7. In Vivo Experiments

Twenty-four male Sprague-Dawley rats (3–4 weeks old, 50–90 g) were supplied by the Laboratory Animal Center of Ningxia Medical University (SPF grade, Certification No. 10752309201900022). All animals were housed under controlled conditions (18–22 °C, 40–70% humidity, 12 h light-dark cycle) with free access to food and water. The experimental procedures were approved by the Ethics Committee of Ningxia Medical University (Approval No. IACUC-NYLAC-2019-008).

Rats were randomly divided into four groups (*n* = 6): Control (deionized water); 2,4-D (75 mg/kg); LBP (50 mg/kg); and 2,4-D + LBP (75 mg/kg + 50 mg/kg) [6], treatments were administered by daily oral gavage for 28 consecutive days. 2,4-D (≥99% purity) was purchased from Sigma-Aldrich, and LBP (≥80% purity) was likewise for the following company: Shanghai Yuanye Bio-Technology, Shanghai, China.

General behavior (e.g., upright posture, scratching, escape, and avoidance responses) and morphological features (skin, mucosa, and fur condition) were monitored daily. Body weight was recorded every two days. After the treatment period, all rats were euthanized. Blood samples (≈10 mL/rat) were collected, and liver and kidney tissues were excised and weighed. Organ coefficient (%) = [organ weight (g)/body weight (g)] × 100.

### 5.8. HE Staining of Liver and Kidney Tissue Samples

Liver and kidney tissues were immediately washed in saline, fixed in 10% neutral buffered formalin, and processed through standard dehydration, paraffin embedding, and sectioning. Sections were stained with H&E for histopathological examination using an phase-contrast microscope (Olympus, Tokyo, Japan).

### 5.9. Determination of SOD, GSH-Px, and MDA in the Serum

Serum was isolated by centrifugation (10,000× *g*, 10 min, 4 °C). MDA, SOD, and GSH-Px levels were quantified using commercial kits from from Nanjing Jiancheng Bioengineering Institute (Nanjing, China) for MDA and GSH-Px, and from Thermo Fisher Scientific (Frederick, MA, USA) for SOD, following the manufacturers’ protocols.

### 5.10. Assessment of Apoptosis Using TUNE L Fluorescence

Liver and kidney sections were subjected to antigen retrieval in citrate buffer (pH 6.0 and 3 × 5 min) after PBS washing, then cooled to room temperature (24 ± 1 °C). Sections were incubated with 50 µL TUNEL reaction mixture (Wanlei Bio, Shenyang, China) at 37 °C for 90 min, followed by DAPI counterstaining. After sealing with anti-fade mounting medium, apoptotic cells were imaged under a Nikon Eclipse Ci-L fluorescence microscope. Apoptotic rates were quantified based on three random fields per sample using the formula: (apoptotic cells/total cells) × 100%.

### 5.11. Statistical Methods

Data are presented as mean ± SD. Statistical analyses were performed using SPSS 25.0. After verifying homogeneity of variance, one-way ANOVA was applied, followed by the SNK-q test (for homogeneous data) or nonparametric tests. A value of *p* < 0.05 was considered statistically significant.

## 6. Conclusions

This study applies network toxicology to elucidate the mechanisms of pesticide-induced organ damage, moving beyond traditional single-mechanism research. On one hand, the PPI network identifies the PPARG-NFKB1-NFE2L2 target cluster, providing a novel combination of biomarkers for 2,4-D toxicity; on the other hand, it experimentally confirmed that exposure to 2,4-D can cause liver and kidney tissue damage in rats by inducing oxidative stress and cell apoptosis, manifested as hepatocyte swelling, vacuolisation, and inflammatory infiltration, pathological changes such as glomerular structural damage and renal tubular epithelial degeneration, accompanied by a significant decrease in SOD and GSH-Px activity and an increase in MDA levels. LBP administration significantly mitigated the aforementioned pathophysiological alterations and oxidative stress, demonstrating substantial protective efficacy against 2,4-D-induced hepatorenal injury. Through its multi-target mechanism, LBP exerts protective effects likely through modulation of the oxidative stress and apoptosis network associated with these key targets.

## Figures and Tables

**Figure 1 ijms-26-10685-f001:**
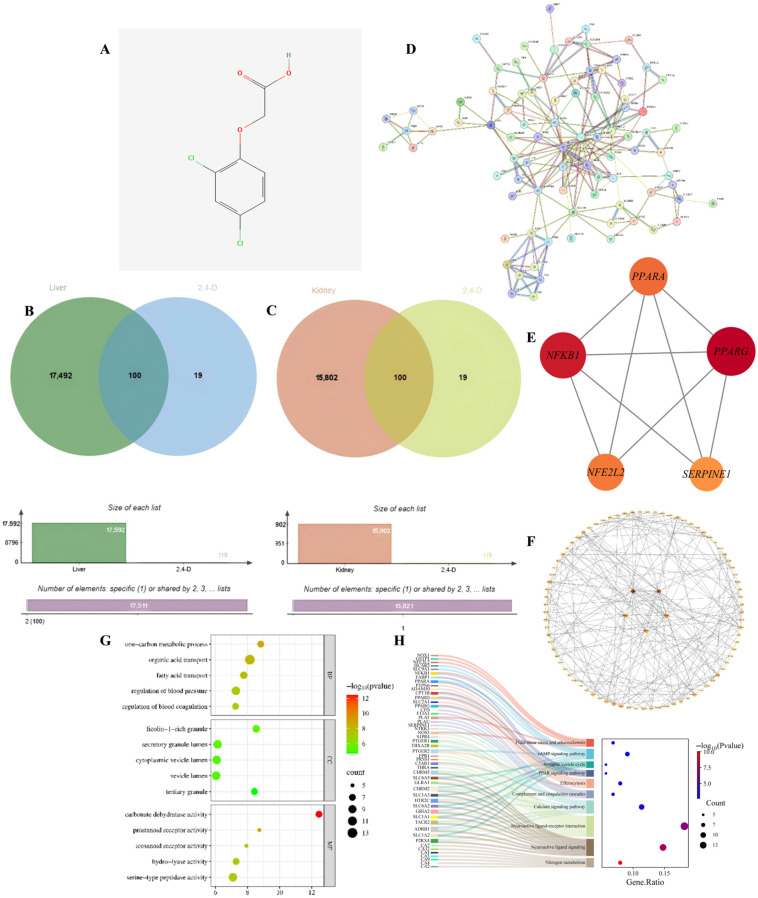
(**A**) Two-dimensional chemical structure of 2,4-D; (**B**) Venn diagram illustrating common targets between 2,4-D and liver apoptosis-related genes; (**C**) Overlap of target genes associated with 2,4-D and kidney apoptosis; (**D**) PPI network constructed for apoptosis-related targets in liver and kidney tissues; (**E**) PPI network diagram of core targets of liver and kidney apoptosis (circles); (**F**) The top five hub nodes with the highest degree of liver and kidney apoptosis were identified and are indicated by redder node colours in the figure; (**G**) Hepatic and renal apoptosis GO analysis bubble chart (BP, CC, MF); (**H**) KEGG enrichment results showing the relationship between target genes and the top 10 pathways in order of hepatic and renal apoptosis gene proportion.

**Figure 2 ijms-26-10685-f002:**
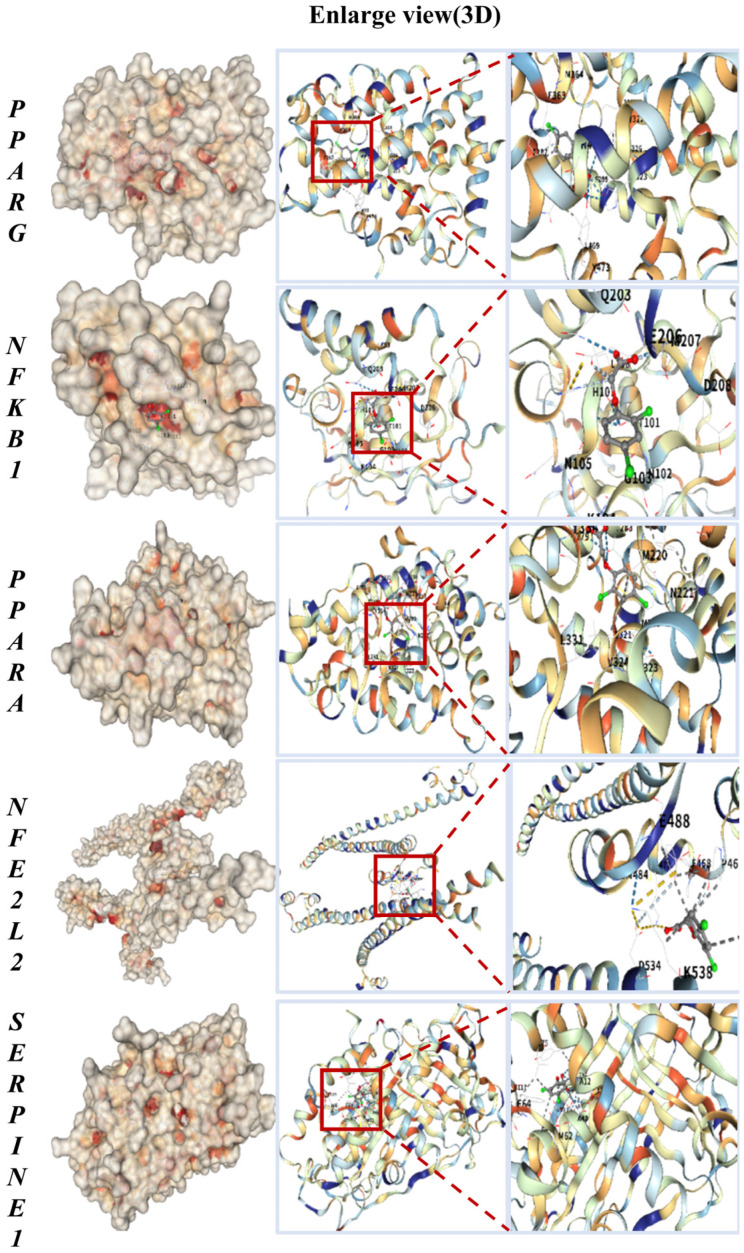
Molecular docking results of 2,4-D with *PPARG*, *NFKB1*, *PPARA*, *NFE2L2*, and *SERPINE1*, all showing the lowest binding energy.

**Figure 3 ijms-26-10685-f003:**
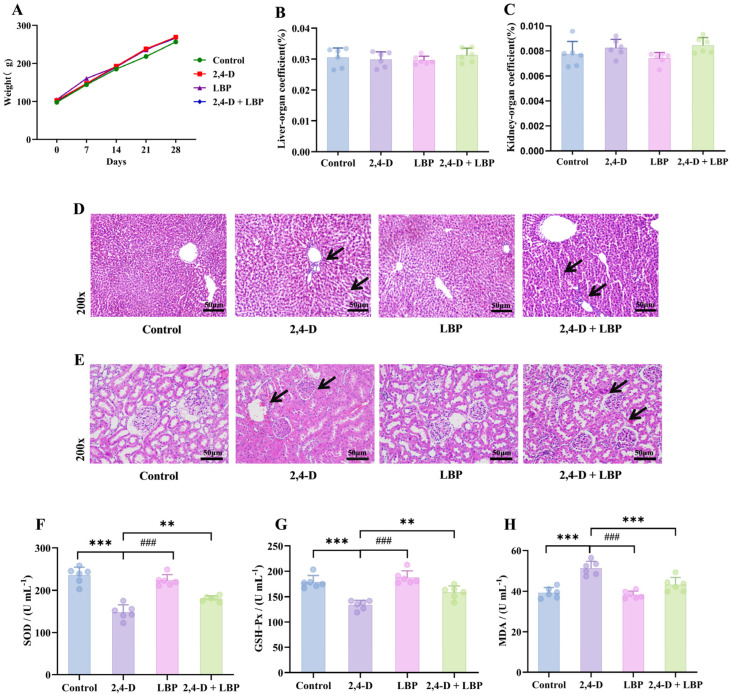
Overall health record of the experimental rats. (**A**) Body weight changes over the 28-day experimental period. The colors represent the different treatment groups: Green, Control (deionized water); Red, 2,4-D (75 mg/kg); Purple, LBP (50 mg/kg); Blue, 2,4-D + LBP (75 mg/kg + 50 mg/kg). Data are presented as mean ± SD (*n* = 6). No statistically significant differences in body weight were observed among the groups (*p* > 0.05). Organ coefficient of the (**B**) liver and (**C**) kidneys. Values are represented as mean + standard deviation. (**D**) Pathological morphologic changes in the rat liver tissues upon 2,4-D exposure and LBP treatment. Liver tissue samples from different experimental groups were stained with hematoxylin-eosin (magnification, 200×). (**E**) Pathological morphologic changes in the rat kidney tissues upon 2,4-D exposure and LBP treatment. Kidney tissue samples from different experimental groups were stained with hematoxylin-eosin (magnification, 200×). Comparison of the serum concentrations of (**F**) SOD, (**G**) GSH-Px, and (**H**) MDA in each group (*** *p* < 0.001 vs. control group; ^###^ *p* < 0.001 & ** *p* < 0.01 vs. the 2,4-D group; Note: (**H**) 2,4-D vs. 2,4-D + LBP-*** *p* < 0.001). ($\bar{x}\pm s$, *n* = 6).

**Figure 4 ijms-26-10685-f004:**
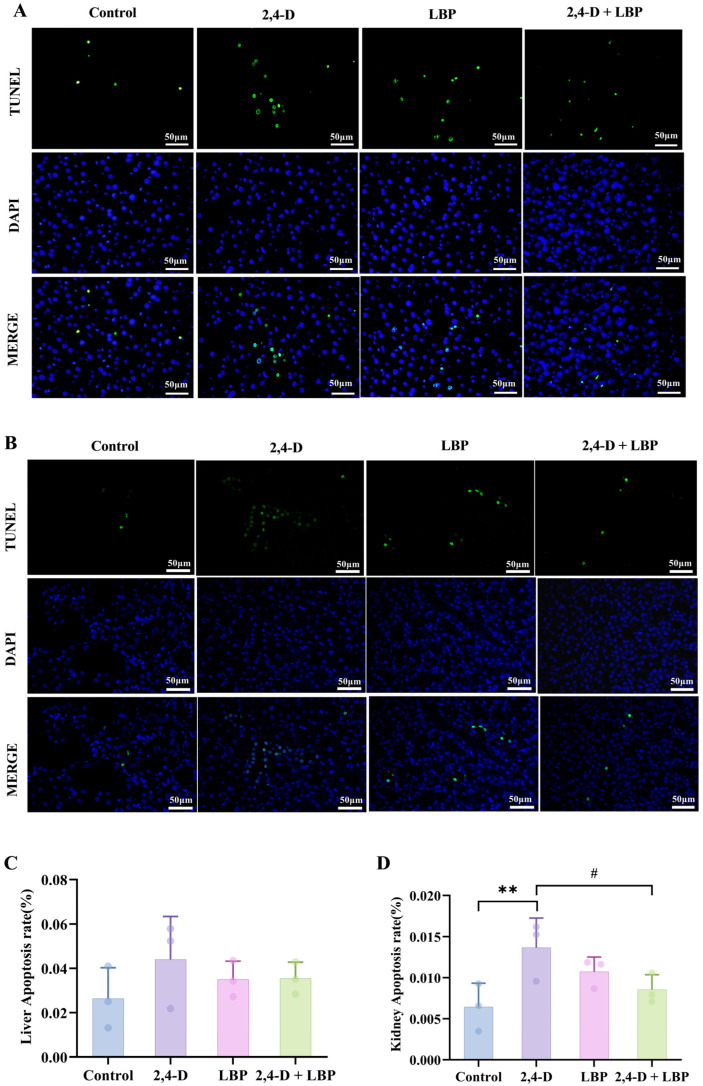
(**A**) Liver and (**B**) kidney cell apoptosis was measured through TUNEL staining. Apoptotic cells are represented in green, and cell nuclei are represented in blue. Comparison of the apoptosis rate in the (**C**) Liver and (**D**) Kidney cells in each group. ** *p* < 0.01 vs. control group. ^#^ *p* < 0.05 vs. the 2,4-D group. ($\bar{x}\pm s$, *n* = 3).

**Table 1 ijms-26-10685-t001:** Molecular docking results of 2,4-D with the core targets and RMSD values from protocol validation.

Ingredient	Target	PDB ID	Unique Ligands	Binding Energy (kcal·mol^−1^)	Validation RMSD (Å)
2,4-D	*PPARG*	8B8X	A8R	−5.9	0.071
2,4-D	*NFKB1*	8TQD	JMR	−5.1	0.066
2,4-D	*PPARA*	8YWV	A1L0A	−6.3	0.065
2,4-D	*NFE2L2*	7X5F	MAFG	−5.2	0.066
2,4-D	*SERPINE1*	3CVM	NONE	−6.0	0.065

## Data Availability

The original contributions presented in this study are included in the article. Further inquiries can be directed to the corresponding author(s).

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
