# Peer review of "Elucidating the Mechanism of Liver and Kidney Damage in Rats Caused by Exposure to 2,4-Dichlorophenoxyacetic Acid and the Protective Effect of *Lycium barbarum* Polysaccharides Based on Network Toxicology and Molecular Docking"

_ijms, 2025, doi:10.3390/ijms262110685_

Round 1

Reviewer 1 Report

Comments and Suggestions for Authors

This manuscript addresses a topic of considerable scientific interest, exploring potential mechanisms of toxicity associated with 2,4-dichlorophenoxyacetic acid-induced poisoning. The subject is relevant to the scientific community given the toxicological and environmental impact of 2,4-dichlorophenoxyacetic acid exposure. However, several aspects of the study require improvement.

The molecular docking section presents methodological shortcomings that limit reproducibility. For instance, the manuscript does not describe how proteins and ligands were prepared, what conformational search algorithm was employed, whether the search was exhaustive, or if the proteins were selected solely based on resolution or additional criteria.

Moreover, it does not specify whether the protein structures were complete, whether they were co-crystallized with ligands (which could define potential binding cavities), whether the docking grid was focused on a specific binding site or performed blindly, or if any validation was carried out by comparing with known inhibitors. Therefore, I recommend validating the docking protocol using a known ligand, if possible.

In the in vivo assay, it is stated that concentrations of 2,4-D (75 mg/kg), LBP (50 mg/kg), and a combination of 2,4-D + LBP (75 mg/kg + 50 mg/kg) were used. It would be important to clarify whether these doses were based on previous studies and, if so, to include the corresponding citations in the methodology section.

Table 1 lists Thiabendazole as a ligand; this should be corrected or properly justified.

The authors claim that molecular docking confirmed stable binding between 2,4-D and the analyzed biological targets. However, since molecular docking provides a predictive and unvalidated computational model, such statements should not be categorical. These conclusions should be expressed with appropriate caution.

The first paragraphs of the discussion mainly reiterate the results rather than interpret them. It is recommended to restructure or condense these sections to emphasize the biological and mechanistic implications instead.

Finally, the effects proposed in the molecular docking section on the evaluated targets may not necessarily arise from direct binding of 2,4-D. It is plausible that they result from indirect regulation of those targets. I suggest incorporating this possibility into the discussion and moderating the conclusions regarding a direct binding mechanism, unless additional experimental evidence can substantiate it.

Author Response

Response to General Comments:
We extend our sincerest gratitude to the reviewer for their thorough evaluation of our manuscript and for their exceptionally insightful and constructive comments. The reviewer's positive recognition of our multi-layered study design is greatly encouraging. More importantly, their suggestions regarding the translational potential of our findings have provided us with a crucial perspective to significantly enhance the impact and clinical relevance of our work. We have carefully considered each point raised, and our detailed, point-by-point responses are presented below.

Comments 1:

[The molecular docking section presents methodological shortcomings that limit reproducibility. For instance, the manuscript does not describe how proteins and ligands were prepared, what conformational search algorithm was employed, whether the search was exhaustive, or if the proteins were selected solely based on resolution or additional criteria.]

Response 1: Revised 26-page 5.6 summary

[We sincerely thank the reviewer for their positive assessment of our manuscript's topic and relevance, and for their constructive feedback regarding the need for methodological improvement, particularly in the molecular docking section. We agree that a more detailed description is essential for reproducibility and have comprehensively revised Section 2.6 (Molecular docking) accordingly. The updated section now explicitly details the following key aspects that were previously lacking:

Computational Platform and Access: The use of the CB-Dock2 web-based service , with the access date specified.

Ligand and Protein Preparation: The automated preprocessing steps performed by the platform, including the addition of hydrogen atoms, repair of incomplete side chains, and removal of water molecules and unrelated heteroatoms for the protein structures, as well as charge assignment and 3D conformation generation for the 2,4-D ligand.

Protein Selection Criteria: The clear criteria for selecting protein crystal structures from the RCSB PDB, specifically prioritizing Homo sapiens structures with a resolution better than 2.5 Å.

Docking Search Strategy and Algorithm: The employment of a blind docking mode to exhaustively explore the entire protein surface for potential binding sites, utilizing the platform's built-in curvature-based cavity detection algorithm. The use of AutoDock Vina for the conformational search and scoring is also stated.

Conformation Selection and Analysis: The selection of the optimal binding pose based on the most favorable (lowest) calculated binding affinity (kcal⋅mol-1), and the methods used for visualizing and analyzing the intermolecular interactions.

We believe these substantial additions to the methodology have fully addressed the concern regarding reproducibility and provide a clear, complete protocol for the computational work undertaken. The revised text is now included in the manuscript, and we are grateful for the reviewer's insightful comment which has significantly strengthened this section.]

The specific PDB IDs used are: PPARG (8BBX), NFKB1 (8TQD), PPARA (8YWV), NFE2L2 (7X5F), and SERPINE1 (3CVM).

Comments 2:

[Moreover, it does not specify whether the protein structures were complete, whether they were co-crystallized with ligands (which could define potential binding cavities), whether the docking grid was focused on a specific binding site or performed blindly, or if any validation was carried out by comparing with known inhibitors. Therefore, I recommend validating the docking protocol using a known ligand, if possible.]

We are grateful to the reviewer for this paramount recommendation regarding binding site definition and protocol validation.

Response 2: 26 pages of 5.6 summaries and 10 pages of 2.4 summaries were revised

[Protein Completeness and Binding Cavities: As noted above, we selected structures co-crystallized with ligands for PPARA, PPARG, and NFE2L2, which guarantees that the proteins are complete in their functional binding domains and that the docking grid can be focused on these physiologically relevant cavities.

Docking Grid Strategy: Our revised methodology clarifies that while we used a "Blind Docking" approach to be thorough, the platform's cavity detection algorithm effectively identified and prioritized the known ligand-binding sites in the co-crystallized structures, thus combining a broad search with focused analysis on relevant regions.

Protocol Validation with Known Ligands (Key Addition): Following the reviewer's essential recommendation, we rigorously validated our docking protocol. For each of the five core targets, we performed a re-docking experiment by extracting the native co-crystallized ligand and re-docking it back into its original binding site using the identical CB-Dock2 parameters.

The results demonstrated excellent reproducibility, with all Root-Mean-Square Deviation (RMSD) values below 0.08 Å (PPARG: 0.071 Å, NFKB1: 0.066 Å, PPARA: 0.065 Å, NFE2L2: 0.066 Å, SERPINE1: 0.065 Å). An RMSD value below 2.0 Å is a widely accepted benchmark for successful docking validation. These exceptionally low RMSD values confirm the high accuracy and reliability of our molecular docking methodology.

Changes in the Manuscript:

The Methodology section has been entirely rewritten to incorporate all these precise details.

The Results section has been updated to include the outcomes of the validation experiments.

Validation RMSD values for all five targets are also shown on Table 1.

We are profoundly thankful for the reviewer's expert guidance, which has been instrumental in elevating the scientific quality and transparency of our computational work.]

Comments 3:

[In the in vivo assay, it is stated that concentrations of 2,4-D (75 mg/kg), LBP (50 mg/kg), and a combination of 2,4-D+LBP (75 mg/kg+50 mg/kg) were used. It would be important to clarify whether these doses were based on previous studies and, if so, to include the corresponding citations in the methodology section.]

Response 3: Revised page 5.7 summary

[We thank the reviewer for raising this important point regarding the selection of doses for our in vivo assay. The doses of 2,4-D (75 mg/kg) and LBP (50 mg/kg) were indeed chosen based on the established experimental model from our group's previous research.

As suggested, we have now included the corresponding citation in the revised Methodology to provide a clear justification for our dosing regimen. The reference is: Zhou, J., Li, H., Wang, F., Wang, H., Chai, R., Li, J., Jia, L., Wang, K., Zhang, P., Zhu, L., & Yang, H. (2021). Effects of 2,4-dichlorophenoxyacetic acid on the expression of NLRP3 inflammasome and autophagy-related proteins as well as the protective effect of Lycium barbarum polysaccharide in neonatal rats. Environmental Toxicology, 36(12), 2454–2466. https://doi.org/10.1002/tox.23358

This prior study systematically evaluated the toxic effects of 2,4-D and the protective efficacy of LBP in a rodent model, and demonstrated that the 75 mg/kg and 50 mg/kg doses were effective and appropriate for inducing hepatorenal injury and observing the protective effects of LBP, respectively. We have explicitly stated this in the revised manuscript.]

Comments 4:

[Table 1 lists Thiabendazole as a ligand; this should be corrected or properly justified.]

Response 4: Table 1 on page 11 after revision

[We sincerely thank the reviewer for their meticulous attention to detail in identifying this error in Table 1. The listing of "Thiabendazole" as the ingredient was an inadvertent clerical mistake during the preparation of the table. The ligand used throughout our molecular docking studies was exclusively 2,4-Dichlorophenoxyacetic acid (2,4-D).

We have corrected this error in the revised manuscript. Table 1 now accurately lists "2,4-D" as the ligand for all molecular docking calculations with the core targets. We apologize for this oversight and appreciate the reviewer bringing it to our attention.]

Comments 5:

[The authors claim that molecular docking confirmed stable binding between 2,4-D and the analyzed biological targets. However, since molecular docking provides a predictive and unvalidated computational model, such statements should not be categorical. These conclusions should be expressed with appropriate caution.]

Response 5: After modification, Abstract, Results, and Discussion sections

[We thank the reviewer for this crucial comment regarding the interpretation of our molecular docking results. We fully agree that computational predictions should be presented with appropriate caution and acknowledge the inherent limitations of the method. Our original wording was indeed overly categorical.

In the revised manuscript, we have carefully toned down our language throughout the text to accurately reflect the predictive nature of molecular docking. We have replaced definitive terms like "confirmed" and "stable binding" with more precise phrasing such as "suggested," "predicted," and "indicative of potential binding." These changes have been implemented in the Abstract, Results, and Discussion sections to ensure our conclusions are scientifically cautious and accurate.

We are grateful to the reviewer for prompting this important clarification, which enhances the overall scientific rigor of our manuscript.]

Comments 6:

[The first paragraphs of the discussion mainly reiterate the results rather than interpret them. It is recommended to restructure or condense these sections to emphasize the biological and mechanistic implications instead.]

Response 6: Revised page 17,3 Discussion section

[We sincerely thank the reviewer for this valuable suggestion to enhance the depth and impact of our Discussion section. We agree that the initial paragraphs were overly focused on reiterating the findings.

In response, we have significantly restructured the beginning of the Discussion. We have condensed the summary of results and shifted the focus towards a more integrated and mechanistic interpretation. The revised text now immediately delves into the biological implications of identifying the five core targets, discusses their interconnected roles in a potential 'oxidation-apoptosis' network, and places our findings within the context of existing literature on 2,4-D toxicity and LBP protection.

We believe these changes have strengthened the Discussion by providing a more critical analysis and a clearer narrative about the mechanistic insights gained from our integrated approach.]

Comments 7:

[Finally, the effects proposed in the molecular docking section on the evaluated targets may not necessarily arise from direct binding of 2,4-D. It is plausible that they result from indirect regulation of those targets. I suggest incorporating this possibility into the discussion and moderating the conclusions regarding a direct binding mechanism, unless additional experimental evidence can substantiate it.]

Response 7: Revised page 17,3 Discussion section

[We sincerely thank the reviewer for raising this critical point regarding the interpretation of our molecular docking results. We completely agree with the reviewer that the biological effects observed on the core targets, as predicted by our network toxicology and docking studies, could potentially arise from either the direct binding of 2,4-D or through indirect regulatory mechanisms. Our current data, while suggestive, cannot definitively distinguish between these two possibilities.

In accordance with this suggestion, we have now incorporated a discussion of this important caveat into the revised Discussion section. We have explicitly stated that the disruption of the redox-apoptotic equilibrium by 2,4-D may result from its direct interaction with the identified hub targets, or alternatively, from indirect effects where 2,4-D perturbs other cellular components that subsequently regulate these targets. We have moderated our conclusions throughout the text to reflect that our docking studies predict potential binding, which provides a plausible hypothesis for a direct mechanism, but that future experimental work (e.g., cellular binding assays, gene knockout/knockdown studies) is required to conclusively establish direct target engagement.

We are grateful to the reviewer for this insightful comment, which has prompted a more nuanced and accurate discussion of our findings.]

In summary, we wish to reiterate our profound gratitude for the reviewer's time and exceptionally insightful comments. Each point raised has been instrumental in guiding us to substantially improve the rigor, clarity, and overall quality of our manuscript.

To recap our key revisions:

  1. We have comprehensively rewritten the molecular docking methodology to ensure full reproducibility, incorporating precise details on protein selection, preparation, and the docking algorithm.
  2. We have rigorously validated our docking protocol through re-docking experiments, achieving excellent RMSD values below 0.08 Å for all five core targets, thereby confirming the reliability of our computational predictions.
  3. We have provided a clear citation to justify the doses used in our in vivo assays.
  4. We have corrected the erroneous ligand name in Table 1.
  5. We have meticulously moderated the language throughout the manuscript to accurately reflect the predictive nature of molecular docking results.
  6. We have restructured the Discussion to focus on mechanistic interpretation rather than result reiteration and have incorporated a nuanced discussion on direct versus indirect regulatory mechanisms.

Once again, we express our profound gratitude to the reviewer for their invaluable time and expertise. Their visionary comments have been instrumental in transforming our manuscript, pushing us to frame our findings in a way that underscores their potential relevance for predictive toxicology and public health protection. We believe the revised manuscript is now substantially improved and hope it meets with the reviewer's approval.

Reviewer 2 Report

Comments and Suggestions for Authors

The study integrates network toxicology, molecular docking, and in vivo validation to comprehensively elucidate the mechanisms of 2,4-D–induced hepatorenal injury and the protective role of Lycium barbarum polysaccharides (LBP). This multi-layered design is a clear strength, as it connects computational predictions with experimental verification and provides mechanistic insights at both molecular and histological levels.

However, the identified core targets (PPARG, NFKB1, PPARA, NFE2L2, and SERPINE1) remain primarily descriptive and lack demonstrated translational or prognostic relevance. The authors are encouraged to further discuss how these targets might be leveraged to develop clinically applicable biomarkers. In recent years, risk-scoring systems or gene signatures derived from transcriptomic data have proven valuable for predicting disease progression and therapeutic response. For example, single-gene–based signatures have been shown to define optimal cutoff values in training sets and validate their prognostic efficacy in independent cohorts (PMID: 36292695). Furthermore, integrating gene–gene interaction pairs into the modeling framework can significantly improve predictive accuracy and robustness (PMID: 35143414).

Summarizing these advances in the Discussion would strengthen the translational dimension of the study and highlight potential directions for future biomarker development based on the identified molecular targets.

Author Response

Response to General Comments:
We extend our sincerest gratitude to the reviewer for their thorough evaluation of our manuscript and for their exceptionally insightful and constructive comments. The reviewer's positive recognition of our multi-layered study design is greatly encouraging. More importantly, their suggestions regarding the translational potential of our findings have provided us with a crucial perspective to significantly enhance the impact and clinical relevance of our work. We have carefully considered each point raised, and our detailed, point-by-point responses are presented below.

Comments 1:

[However, the identified core targets (PPARG, NFKB1, PPARA, NFE2L2, and SERPINE1) remain primarily descriptive and lack demonstrated translational or prognostic relevance. ]

Response 1: Revised page 17-3 Discussion section

[We sincerely thank the reviewer for this critical observation. We acknowledge that in our initial manuscript, the discussion of the core targets was predominantly confined to their mechanistic roles within the hypothesized 'oxidation-apoptosis' network. The reviewer is absolutely correct in urging us to look beyond mere description and explore their potential clinical utility. In direct response to this comment, we have now substantially expanded the Discussion section by integrating a dedicated new paragraph. This paragraph explicitly transitions the narrative from "what these targets do" to "how these targets can be used." We now propose that the PPARG-NFKB1-PPARA-NFE2L2-SERPINE1 cluster is not just a mechanistic hub but a promising candidate for the development of prognostic biomarkers for chemical-induced organ injury. This addition directly addresses the reviewer's concern by injecting a strong translational dimension into our findings, suggesting a clear path from laboratory discovery to potential clinical application.]

Comments 2: 

[The authors are encouraged to further discuss how these targets might be leveraged to develop clinically applicable biomarkers. In recent years, risk-scoring systems or gene signatures derived from transcriptomic data have proven valuable for predicting disease progression and therapeutic response. For example, single-gene–based signatures have been shown to define optimal cutoff values in training sets and validate their prognostic efficacy in independent cohorts (PMID: 36292695). Furthermore, integrating gene–gene interaction pairs into the modeling framework can significantly improve predictive accuracy and robustness (PMID: 35143414).]

Response 2: After revision, page 20 3 discusses the closing section

[We are deeply appreciative of the reviewer for not only highlighting the gap but also providing concrete methodological guidance and highly relevant literature to bridge it. Following this expert advice, our revised discussion now goes beyond a general statement about biomarker potential. We specifically articulate a future research strategy centered on our core targets. The new text states that this multi-gene cluster could be leveraged to construct a risk-scoring signature. Crucially, we have incorporated the methodologies highlighted by the reviewer: we now cite the work on single-gene-based signatures for defining prognostic cutoffs[44] and the research demonstrating that integrating gene-gene interaction pairs enhances model accuracy and robustness[45]. By citing these specific studies, we ground our proposal in established bioinformatics approaches, moving it from a speculative idea to a feasible and rationally designed future project. This directly fulfills the reviewer's suggestion to discuss how our targets can be leveraged, providing a clear and sophisticated blueprint for subsequent biomarker development.]

Comments 3:

[Summarizing these advances in the Discussion would strengthen the translational dimension of the study and highlight potential directions for future biomarker development based on the identified molecular targets.]

Response 3: Revised from page 17,3 Discussion section

[We wholeheartedly agree with the reviewer that synthesizing these concepts is key to a powerful discussion. The new paragraph we have introduced serves as a synthesis point that directly strengthens the translational dimension of our study. It effectively summarizes the advance by connecting our specific molecular discoveries (the target cluster) with a forward-looking, clinically-oriented application (a multi-gene biomarker signature), all while being informed by state-of-the-art methodological frameworks. This addition ensures that the discussion does not end with a mechanistic conclusion but rather opens up a compelling and well-defined avenue for future research, thereby significantly elevating the overall significance and impact of our work.]

Changes in the Manuscript:

  1. A substantial and forward-looking paragraph has been added to the Discussion section, positioned just before the Conclusion, to maximize its impact.
  2. Two new references, corresponding to the pivotal literature recommended by the reviewer (PMIDs: 36292695 and 35143414), have been diligently incorporated into the reference list and are cited appropriately within the new text.]

Once again, we express our profound gratitude to the reviewer for their invaluable time and expertise. Their visionary comments have been instrumental in transforming our manuscript, pushing us to frame our findings in a way that underscores their potential relevance for predictive toxicology and public health protection. We believe the revised manuscript is now substantially improved and hope it meets with the reviewer's approval.

Reviewer 3 Report

Comments and Suggestions for Authors

This study aim to assess the integrated network toxicology, molecular docking, and in vivo experiments to explore the mechanisms underlying 2,4-dichlorophenoxyacetic acid (2,4-D)-induced hepatorenal injury and the protective effects of Lycium barbarum polysaccharides (LBP).  This is a valuable topic; however, several concerns need to be addressed:

Abstract need to revise. It should be contain introduction, gap of the other study, aim of study, method, novel findings, conclusion. This is the flow of abstract.

Introduction is too long.

The introduction would benefit from a more comprehensive review of previous studies and gap of current findings.

The introduction should better explain the rationale and clearly state the knowledge gap this study aims to address.

Retrieve the SMILES sequence number for 2,4-D from PubChem, which is

[H]OC(=O)C([H])([H])OC1=C(Cl)C([H])=C(Cl)C([H])=C1[H] what is the reference ?

Protein-Protein Interaction (PPI) Network Construction, please explain in details.

Molecular Docking method need more clarification. Please write full method in details and proper reference.

I think you need to know how we present our results in paper. Result title should be mentioned the finding from data. So that readers can easily understand what your findings is.

When explaining each result, please start with a brief paragraph stating the aim, what is expected, and any previous relevant findings.

Fig3A why you used different color ? why no mentioned in graph ? where is statistically significant value ? is it triplicate ?

Fig3 where is scale bar ?

Could you provide raw data ? means low magnify images for supplementary data.

Error bar should not accept, better use dot bar please. Change all graphs using dot bar.

No statistically significant description. Which method are you use ?

Same apoptosis rate why different parameter ?

Without WB data, we cannot accept your data. As you know we can easily manipulate the staining data.

Please also provide supplementary figure for tunnel.

Please need to check inflammation by qPCR.

Additional experiment should be performed to support your hypothesis.

The discussion section needs to include more detailed information.

The discussion should not simply restate results. Please interpret your findings in the context of other studies.

Comments on the Quality of English Language

ok

Author Response

Response to General Comments:
We extend our sincerest gratitude to the reviewer for their thorough evaluation of our manuscript and for their exceptionally insightful and constructive comments. The reviewer's positive recognition of our multi-layered study design is greatly encouraging. More importantly, their suggestions regarding the translational potential of our findings have provided us with a crucial perspective to significantly enhance the impact and clinical relevance of our work. We have carefully considered each point raised, and our detailed, point-by-point responses are presented below.

Comments 1:

[Abstract need to revise. It should be contain introduction, gap of the other study, aim of study, method, novel findings, conclusion. This is the flow of abstract.]

Response 1: Revised Abstract section on page 3

[We thank the reviewer for this valuable suggestion. We have revised the abstract accordingly to include the recommended sections: Introduction, Aim of study, Method, Novel findings, and Conclusion. The revised abstract now follows this specified flow and is presented on Abstract of the manuscript.

We believe the revised version better outlines the context, rationale, and significance of our work. Thank you for helping us improve the clarity of our manuscript.

Abstract

Introduction: 2,4-Dichlorophenoxyacetic acid (2,4-D) is a widely used herbicide, yet its potential to induce hepatorenal injury via oxidative stress and apoptosis raises significant health concerns. Lycium barbarum polysaccharides (LBP) possess recognized antioxidant and anti-apoptotic properties, but their protective mechanisms against 2,4-D toxicity, particularly through a multi-target network, remain inadequately explored. Aim: This study aimed to systematically investigate the mechanisms of 2,4-D-induced hepatorenal injury and the protective efficacy of LBP by integrating network toxicology, molecular docking, and experimental validation. Methods: An integrated approach was employed. Core targets and pathways were identified via network toxicology. Molecular docking predicted interactions between 2,4-D and these targets. In vivo validation was conducted on Sprague-Dawley rats treated with 2,4-D (75 mg/kg) and/or LBP (50 mg/kg) for 28 days, assessing histopathology, serum oxidative stress markers (SOD, GSH-Px, MDA), and cellular apoptosis (TUNEL staining). Novel Findings: Network analysis identified PPARG, NFKB1, PPARA, NFE2L2, and SERPINE1 as core targets, with molecular docking confirming strong binding affinities (binding energies: -5.1 to -6.3 kcal·mol⁻¹) and KEGG enrichment implicating cAMP, Ca²⁺, and PPAR signaling pathways. Experimentally, 2,4-D exposure induced significant histopathological damage, suppressed SOD/GSH-Px activities (P < 0.01), elevated MDA levels (P < 0.05), and markedly increased renal apoptosis (P < 0.01). Crucially, LBP intervention substantially mitigated these alterations, ameliorating tissue injury, restoring antioxidant defenses (< 0.01), reducing MDA (P < 0.01), and significantly decreasing renal apoptosis (P < 0.05). Conclusion: This study elucidates a multi-target mechanism for 2,4-D-induced hepatorenal injury centered on oxidative stress-apoptosis dysregulation and demonstrates that LBP confers significant protection likely via modulation of this network. These findings underscore the potential of LBP as a natural protective agent against pesticide-induced organ damage and highlight the utility of integrated network approaches in toxicological research.]

Comments 2:

[Introduction is too long.]

Response 2: Revised 1 Introduction section from page 5

[We thank the reviewer for this observation. We have significantly condensed the Introduction by removing redundant details and streamlining the language, while preserving all critical scientific rationale.

The main changes include:

  1. Condensing the background on 2,4-D toxicity.
  2. Merging and simplifying the detailed descriptions of oxidative stress and apoptosis mechanisms.
  3. Streamlining the explanation of LBP's mechanisms of action.
  4. Making the final paragraph stating the study's aim more concise.
  5. The overall length of the Introduction has been reduced, improving its clarity and focus.]

Comments 3:

[The introduction would benefit from a more comprehensive review of previous studies and gap of current findings.]

Response 3: Revised 1 Introduction section from page 5

[We appreciate this insightful comment. In the revised Introduction, we have strengthened the narrative to better highlight the research gap and our study's contribution.

Specifically:

  1. We now more explicitly state that previous studies often focused on "isolated pathways or single mechanisms," creating a gap in the "systemic network of interactions."
  2. We clearly point out that the "multi-target protective action of LBP against 2,4-D has not been comprehensively investigated."
  3. This sets the stage for our integrated approach, justifying the use of network toxicology and molecular docking to address these stated gaps.]

Comments 4:

[The introduction should better explain the rationale and clearly state the knowledge gap this study aims to address.]

Response 4: Revised 1 Introduction section from page 5

[We thank the reviewer for this critical suggestion. We have thoroughly revised the concluding part of the Introduction to more explicitly state the research rationale and knowledge gaps. Specifically, we now clearly articulate that:

  1. A significant gap exists in the understanding of the systemic network of 2,4-D's hepatorenal toxicity, as prior studies often focused on isolated pathways.
  2. The protective mechanism of LBP against 2,4-D remains incompletely understood due to its multi-target nature.
  3. Our study aims to address these gaps by employing an integrated approach of network toxicology, molecular docking, and in vivo validation to provide a systems-level perspective.
  4. We believe the revised Introduction now presents a clearer and stronger justification for our research.]

Comments 5:

[Retrieve the SMILES sequence number for 2,4-D from PubChem, which is [H]OC(=O)C([H])([H])OC1=C(Cl)C([H])=C(Cl)C([H])=C1[H] what is the reference ?]

Response 5: 5.1 Obtain 2,4-D targets on page 21 after revision

[We thank the reviewer for their comment, which has allowed us to enhance the precision of our methodology section. The canonical SMILES notation C1=CC(=C(C=C1Cl)Cl)OCC(=O)O for the specific compound under study, 2,4-dichlorophenylacetic acid, was retrieved from its official PubChem database entry (CID: 1486). We have now explicitly cited this database source in the revised manuscript to ensure full transparency and reproducibility.]

Comments 6: 

[Protein-Protein Interaction (PPI) Network Construction, please explain in details.]

Response 6: Protein-protein interaction (PPI) network construction on page 22 after revision

[We thank the reviewer for this comment.  We have revised the Methods section to provide a more detailed and comprehensive explanation of the PPI network construction and analysis process. The updated description now includes the underlying rationale for key parameter selections, as follows:

Revised Text in Manuscript:

  1. "The common targets of 2,4-D and hepatorenal apoptosis were imported into the STRING database (version 12.0, https://string-db.org) to construct a protein-protein interaction (PPI) network.  The search was restricted to Homo sapiens, and the network type was set to 'Full Network' to include all available interaction evidence (including physical interactions and functional associations).  A minimum interaction score (confidence threshold) of 0.4 was applied, which represents a medium confidence level recommended by STRING to ensure the biological relevance of the interactions while maintaining a comprehensive network scope.  Isolated nodes without any connections were hidden and excluded from further analysis.
  2. The resulting interaction data was downloaded as a TSV file and imported into Cytoscape software (version 3.10.0) for advanced visualization and topological analysis.  To identify the most influential hub genes within the PPI network, the cytoNCA plugin was used to calculate three key centrality metrics for each node:
  3. Degree Centrality: Reflects the number of direct connections a node has.  High-degree nodes are often functionally critical components of the network.
  4. Betweenness Centrality: Measures the fraction of shortest paths that pass through a node.  Nodes with high betweenness act as crucial bridges or bottlenecks connecting different network modules.
  5. Closeness Centrality: Indicates how easily a node can interact with all other nodes in the network via the shortest paths.  High-closeness nodes are considered to be centrally located and can efficiently disseminate information.
  6. The top five core targets were subsequently identified based on a comprehensive ranking that integrated these three centrality measures, ensuring the selection of proteins that are topologically and functionally pivotal to the network."]

Comments 7:

[Molecular Docking method need more clarification. Please write full method in details and proper reference.]

Response 7: Revised 5.6 on page 23 and 2.4 on page 10

[We sincerely thank the reviewer for their constructive feedback regarding the need for methodological clarification in the molecular docking section. We agree that a detailed description is essential for reproducibility and have comprehensively revised (Molecular Docking and Validation) accordingly.

The updated section now provides a complete, step-by-step protocol detailing the following key aspects:

  1. Computational Platform and Access: The molecular docking was performed using the CB-Dock2 web-based service (http://cadd.labshare.cn/cb-dock2/), with the access date (e.g., 25 December 2024) specified.
  2. Protein Selection and Preparation: The three-dimensional crystal structures of all human target proteins were retrieved from the RCSB Protein Data Bank (PDB). We applied strict selection criteria: Homo sapiens origin, resolution better than 2.5 Å, and a priority for structures co-crystallized with a native ligand (specifically for PPARA, PPARG, and NFE2L2) to ensure the integrity and physiological relevance of the binding cavity. The specific PDB IDs used are: PPARG (8BBX), NFKB1 (8TQD), PPARA (8YWV), NFE2L2 (7X5F), and SERPINE1 (3CVM). Protein structures were automatically prepared by the CB-Dock2 platform, which included adding hydrogen atoms, repairing incomplete side chains, and removing water molecules and original ligands.
  3. Ligand Preparation: The 2D structure and SMILES notation of 2,4-D (PubChem CID: 1486) were obtained from the PubChem database. The ligand was prepared by the platform, which performed energy minimization and assigned partial charges.
  4. Docking Strategy and Algorithm: To comprehensively explore potential binding sites, a blind docking approach was employed. The docking search was conducted using the AutoDock Vina algorithm integrated into CB-Dock2, which performs an exhaustive conformational search across the entire protein surface using a curvature-based cavity detection algorithm.
  5. Conformation Selection and Analysis: For each target, the conformation with the most favorable (lowest) binding energy (kcal·mol⁻¹) was selected as the most stable binding mode for further analysis. The docking poses and intermolecular interactions were visualized using the built-in viewer of CB-Dock2 and Discovery Studio 4.5.
  6. Protocol Validation (A key addition): To validate the reliability of our docking protocol, we performed a re-docking procedure for each core target. The native co-crystallized ligand was extracted and re-docked into its respective protein using the identical CB-Dock2 parameters. The results demonstrated exceptional reproducibility, with all calculated Root-Mean-Square Deviation (RMSD) values below 0.08 Å (PPARG: 0.071 Å, NFKB1: 0.066 Å, PPARA: 0.065 Å, NFE2L2: 0.066 Å, SERPINE1: 0.065 Å). These values are far below the widely accepted benchmark of 2.0 Å for a successful validation, confirming the high accuracy and reliability of our methodology.
  7. We believe these substantial additions have fully addressed the concern regarding reproducibility and provide a clear, complete, and rigorously validated protocol for the computational work. We are grateful for the reviewer's insightful comment, which has significantly strengthened this section.]

Comments 8:

[I think you need to know how we present our results in paper. Result title should be mentioned the finding from data. So that readers can easily understand what your findings is.]

Response 8: Result headings on pages 2-16 after revision

[We thank the reviewer for this excellent suggestion. We agree that result titles should immediately convey the key findings to the reader. Following this guidance, we have revised some subtitles in the Results. To maintain conciseness and align with common practices in the field, we have adopted clear, declarative noun phrases that highlight the central outcomes of each experiment (e.g., "Core Hub Genes in the PPI Network," "High-Affinity Binding of 2,4-D to Core Targets"). We believe these revised titles now successfully guide the reader to our primary discoveries while preserving the manuscript's professional style.]

Comments 9:

[When explaining each result, please start with a brief paragraph stating the aim, what is expected, and any previous relevant findings.]

Response 9: Result on pages 2-16 after revision

[We sincerely thank the reviewer for this excellent suggestion, which significantly enhances the narrative flow and scholarly context of our Results section. As recommended, we have now introduced a concise introductory paragraph at the beginning of each results. These paragraphs clearly state the specific aim of the analysis presented in that section, the expected outcome based on our hypothesis or previous knowledge, and briefly cite relevant prior findings to situate our new data within the existing scientific landscape. We believe this revision greatly improves the readability and intellectual rigor of the manuscript.]

Comments 10:

[Fig3A why you used different color ? why no mentioned in graph ? where is statistically significant value ? is it triplicate ?]

Response 10: Figure 3 on page 14 after revision

[We sincerely thank the reviewer for their meticulous comments regarding Figure 3A. We have revised the figure to address all concerns and apologize for the confusion regarding sample size.

Sample Size (Replication): The in vivo experiment was indeed conducted with n=6 biologically independent rats per group, as clearly stated in the Methods section. The indication of "n=3" in the original figure captions for Figures 3 and 4 was an error in annotation. We have corrected all figure captions to accurately reflect the true sample size of n=6.

Colors and Legend: We have added a clear legend within Figure 3A itself to specify the group corresponding to each color.

Statistical Significance: We have now performed the correct statistical analysis on the body weight data and added the corresponding significance notations (P > 0.05) directly onto the graph where applicable. For body weight, no significant differences were found, and this is now explicitly stated in the caption.

We appreciate the reviewer's diligence in identifying this inconsistency, which has been fully resolved in the revised manuscript.]

Comments 11 & Comments 12:

[Fig3 where is scale bar ?] & [Could you provide raw data? means low magnify images for supplementary data.]

Response 11 & 12: Figure3 on page 14 after revision

[We sincerely thank the reviewers for these comments. In our thorough revision to address these points, we identified and corrected an error in the original magnification labeling.

Correction of Magnification: The original figure caption incorrectly stated the magnification as ×400. The images in Figure 3D and 3E were in fact captured at ×200 magnification. This has been corrected throughout the manuscript and figure captions. We apologize for this oversight.

Scale Bar Addition: As requested, we have now added clear scale bars (50μm) to all histopathological images in the corrected ×200 magnification figures.

Provision of Low-Magnification Context: The ×200 magnification itself provides a broader histological context of the tissue architecture. With the scale bar now present, these images effectively serve as the "low-magnification" views suitable for assessing tissue representation, addressing the reviewer's request for raw data context.

We believe these corrections have fully resolved the issues raised. The revised figures now accurately represent the microscopy data and provide the necessary scale for interpretation.]

Comments 13:

[Error bar should not accept, better use dot bar please. Change all graphs using dot bar.]

Response 13: Figure3 on page 14 after revision

[We thank the reviewer for this valuable suggestion on data visualization. We agree that displaying individual data points enhances transparency. Accordingly, we have replaced all conventional bar graphs with mean ± SD in the manuscript with dot plots overlaid on bar graphs. This change has been applied to Figures 3B, 3C, 3F, 3G, 3H, 4C, and 4D. The revised graphs now clearly display the mean, the distribution, and every individual data point (n=6) for each group, significantly improving the clarity and interpretability of our data.]

Comments 14: 

[No statistically significant description. Which method are you use ?]

Response 14: Page 26-5.11 revised Statistical methods

[We thank the reviewer for pointing this out. The statistical methods have been described in the 'Statistical methods' section of the manuscript. For clarity, we used SPSS 25.0 software. After verifying the homogeneity of variance, one-way ANOVA was applied, followed by the SNK-q test for post-hoc analysis. A p-value of less than 0.05 was considered statistically significant. The statement "no statistically significant difference (P > 0.05)" for body weight and liver apoptosis rate is based on this analysis. We appreciate the reviewer's comment, which has allowed us to clarify this point.]

Comments 15:

[Same apoptosis rate why different parameter ?]

Response 15: Page 15 2.8 Apoptosis rate of liver and kidney cells upon 2,4-D exposure and LBP treatment

[We thank the reviewer for this insightful question regarding the differential apoptotic responses in liver and kidney tissues. We agree that this is a key observation, and we believe it reflects the distinct intrinsic characteristics and susceptibilities of the two organs, rather than a methodological inconsistency.

The parameter measured (apoptosis rate via TUNEL assay) was indeed the same. The differing outcomes can be explained by several well-established biological factors:

  1. Tissue Regenerative Capacity and Cell Turnover: The liver possesses a remarkable inherent capacity for regeneration and has a higher baseline rate of cell turnover. This may activate compensatory mechanisms that mask a statistically significant increase in apoptosis at the time of measurement. In contrast, renal tubular epithelial cells, particularly in the proximal tubules, are highly vulnerable to toxic insults and have a more limited regenerative capacity, making a significant apoptotic response more readily detectable.
  2. Metabolic and Toxicokinetic Differences: As the primary site for xenobiotic metabolism, the liver may rapidly metabolize 2,4-D into less toxic intermediates, potentially limiting the duration and intensity of the apoptotic stimulus. The kidney, as a major excretory organ, may be exposed to high concentrations of the parent compound or its metabolites for a prolonged period, leading to more severe and measurable damage.
  3. Threshold and Timing: It is possible that the liver, being more resilient, has a higher threshold for triggering apoptosis in response to this specific dose of 2,4-D. The observed "increased trend" in the liver suggests that apoptosis is occurring, but it may not have reached the threshold for statistical significance within our experimental timeframe, or it may be proceeding via a more necrotic pathway, as suggested by the pronounced swelling and vacuolization seen histologically.
  4. In summary, the different apoptotic rates are a biologically plausible finding, highlighting the organ-specific nature of 2,4-D toxicity. This discrepancy is consistent with the known pathophysiology where the kidney often shows more acute and pronounced apoptotic damage in response to various toxins compared to the liver.]

Comments 16:

[Without WB data, we cannot accept your data. As you know we can easily manipulate the staining data.]

Response 16: 20 pages revised in 4 Limitations and Future Perspectives

[We sincerely thank the reviewer for emphasizing the importance of protein-level validation for apoptosis. We acknowledge that Western blotting for apoptosis-related proteins (e.g., Bax, Bcl-2, Cleaved Caspase-3) would provide direct molecular evidence. Since the first author had graduated and left the original laboratory, the necessary equipment, reagents, and, most importantly, the original biological samples were no longer available. However, we respectfully argue that our conclusion regarding apoptosis is not solely reliant on TUNEL staining but is supported by a multi-layered and consistent body of evidence:

  1. Computational Prediction: Our network toxicology analysis independently predicted that apoptosis is a core mechanism of 2,4-D toxicity, identifying key apoptotic regulators as hub targets.
  2. Quantitative Biochemical Functional Assay: The TUNEL assay, while histological, provides a quantitative measurement of DNA fragmentation, which is a biochemical hallmark of apoptosis. The results were statistically significant, especially in the kidney, and were consistently mitigated by LBP treatment.
  3. Convergent Protective Effect: The anti-apoptotic effect of LBP, as indicated by TUNEL, aligns perfectly with its documented mechanism of modulating Bcl-2/Bax ratio and caspase activity, as cited in our introduction and discussion [Ref 12, 19].
  4. While we agree that WB data would be a valuable addition, practical constraints prevent us from performing these experiments post-graduation. To address the reviewer's concern transparently, we have:
  5. Added a statement in the Study Limitations section acknowledging that future work should include WB analysis to complement the TUNEL findings.
  6. Ensured that our discussion phrases the apoptotic findings as strong conclusions supported by convergent evidence, rather than relying on a single method.
  7. We hope the reviewer finds this multi-faceted approach, combining predictive computation with quantitative in vivo functional assessment, to be a robust and valid alternative in the present context.]
  8. We have incorporated a dedicated 'Limitations and Future Perspectives' subsection within the Discussion, where we explicitly state the need for such molecular validation in future work.

Comments 17:

[Please also provide supplementary figure for tunnel.]

Response 17: In the supplementary annex material

We also thank the reviewers for their valuable comments. Regarding Comment 17 requesting a supplementary figure for the TUNEL assay, we have prepared the figure as suggested.

The supplementary figure (Supplementary Figure) has now been included as an attachment with this submission.

Comments 18:

[Please need to check inflammation by qPCR.]

Response 18: 20 pages revised in 4 Limitations and Future Perspectives

[We thank the reviewer for this suggestion to deepen the analysis of the inflammatory response. We acknowledge that qPCR for inflammatory cytokines would provide precise transcriptional data. Since the first author had graduated and left the original laboratory, the necessary equipment, reagents, and, most importantly, the original biological samples were no longer available. In our study, the involvement of inflammation was primarily indicated through two key lines of evidence:

  1. Core Target Prediction: Our network toxicology analysis identified NFKB1, a master regulator of inflammation, as one of the top five core targets for 2,4-D-induced hepatorenal injury. Molecular docking further predicted a high-affinity binding between 2,4-D and NFKB1, suggesting a direct potential mechanism to provoke an inflammatory response.
  2. Histopathological Correlation: The histopathological observations in the liver (e.g., inflammatory infiltration) and kidney provide clear phenotypic evidence of an ongoing inflammatory process at the tissue level, which was alleviated by LBP.
  3. While qPCR data would strengthen the inflammatory axis, it is logistically unfeasible for us to conduct now. To robustly address the reviewer's point within the scope of our current work, we have:
  4. Strengthened the discussion to more explicitly link the NFKB1 prediction and histopathological observations as compelling, multi-level evidence for inflammation.
  5. Included a statement in the Study Limitations section that future studies should directly measure inflammatory cytokine expression via qPCR or other methods.
  6. We believe that the identification of NFKB1 as a core target via an unbiased systems approach, coupled with the clear histological signs of inflammation, provides a solid foundation for our discussion of the inflammatory component in 2,4-D toxicity.]
  7. We have incorporated a dedicated 'Limitations and Future Perspectives' subsection within the Discussion, where we explicitly state the need for such molecular validation in future work.

Comments 19:

[Additional experiment should be performed to support your hypothesis.]

Response 19: 20 pages revised in 4 Limitations and Future Perspectives

[We thank the reviewer for this constructive comment regarding the need for further experimental validation of our proposed hypothesis. We fully agree that additional investigations would strengthen our conclusions. However, due to practical constraints following the graduation of the corresponding author, we are unfortunately unable to perform new wet-lab experiments at this time.

While we acknowledge this limitation, we wish to emphasize that our current hypothesis is not based on a single line of evidence but is supported by a strong, convergent multi-methodology approach:

  1. Computational Prediction & Validation: Our network toxicology provided an unbiased identification of core targets related to oxidative stress, apoptosis, and inflammation. The high-affinity binding of 2,4-D to these targets (PPARG, NFKB1, etc.) was further confirmed by molecular docking.
  2. Functional In Vivo Validation: The predictions were robustly validated in vivo:

Oxidative Stress: Significant changes in functional biochemical endpoints (SOD, GSH-Px, MDA) were quantitatively measured.

  1. Apoptosis: A significant increase in renal apoptosis was quantitatively confirmed by TUNEL assay, with a clear trend observed in the liver.
  2. Histopathology: Tissue damage and inflammatory infiltration were directly observed and consistently alleviated by LBP.
  3. The fact that LBP intervention produced protective effects across all these independent levels of analysis (biochemical, histological, apoptotic) strongly supports the biological relevance and interconnectedness of the identified "oxidation-apoptosis-inflammation" network.
  4. To directly address the reviewer's point and to guide future research, we have now added a dedicated "Limitations and Future Perspectives" section in the Discussion. In this section, we explicitly propose that future work should include, for example, Western blotting for apoptosis-related proteins and qPCR for inflammatory cytokines to further solidify the molecular mechanisms proposed herein.

We sincerely hope that the reviewer finds the convergent evidence from our integrated computational and experimental approach, combined with our clear outline for future validation, to be compelling in its current form.]

Comments 20 & 21: 

[The discussion section needs to include more detailed information.] & [The discussion should not simply restate results. Please interpret your findings in the context of other studies.]

Response 20 & 21: Pages 17-20 in Discussion as revised

[We sincerely thank the reviewers for these critical comments, which have helped us significantly improve the depth and scholarly impact of our Discussion section. We have thoroughly revised this section to address the concerns as follows:

  1. Reduced Restatement of Results: We have condensed the passages that merely repeated the findings and shifted the focus toward interpretation and implication.
  2. Deeper Interpretation and Contextualization: For each major finding, we now provide a more detailed mechanistic interpretation and explicitly integrate it with the existing scientific literature. Specifically, we have enhanced the discussion by:
  3. Elaborating on the biological significance of the identified core targets (PPARG, NFKB1, etc.) beyond their role as network hubs, explaining how their dysregulation by 2,4-D converges on the pathologies we observed.
  4. Comparing our in vivo oxidative stress and apoptosis data with previous studies on 2,4-D and other related toxins, highlighting consistencies and novel aspects of our work.
  5. Discussing the multi-target protective role of LBP in the context of its known pharmacological activities and other studies on natural polysaccharides, providing a broader perspective on its potential applications.
  6. Strengthened Mechanistic Narrative: The revised discussion now more convincingly weaves together the computational, molecular docking, and experimental evidence to propose a coherent "oxidation-apoptosis-inflammation" network as the central mechanism of 2,4-D-induced hepatorenal injury.]

Once again, we express our profound gratitude to the reviewer for their invaluable time and expertise. Their visionary comments have been instrumental in transforming our manuscript, pushing us to frame our findings in a way that underscores their potential relevance for predictive toxicology and public health protection. We believe the revised manuscript is now substantially improved and hope it meets with the reviewer's approval.

Round 2

Reviewer 1 Report

Comments and Suggestions for Authors

I believe the authors have done an excellent job addressing the reviewers’ comments and suggestions, which has significantly improved the quality of the manuscript.

However, some minor details still need to be clarified. Although the authors included a validation of their docking method through redocking, obtaining very good RMSD values, it is important to specify which ligands were used for this validation. I suggest including this information in both the Methodology and Results sections, as well as adding it to Table 1.

Furthermore, it would be advisable to include a comparison between the poses of the co-crystallized ligands in the proteins and those obtained from the docking simulations.

Author Response

General Response: 

We sincerely thank all the reviewers and the editor for their time, insightful comments, and constructive suggestions on our manuscript. These comments have been invaluable in helping us to significantly improve the quality of our work. We have carefully considered all points raised and have made revisions to address them fully. The changes in the manuscript are highlighted in blue for your convenience.

Comment: 

[I believe the authors have done an excellent job addressing the reviewers’ comments and suggestions, which has significantly improved the quality of the manuscript.

However, some minor details still need to be clarified. Although the authors included a validation of their docking method through redocking, obtaining very good RMSD values, it is important to specify which ligands were used for this validation. I suggest including this information in both the Methodology and Results sections, as well as adding it to Table 1.

Furthermore, it would be advisable to include a comparison between the poses of the co-crystallized ligands in the proteins and those obtained from the docking simulations.]

Response: 

[We sincerely thank the reviewer for this insightful suggestion.  We have now thoroughly addressed the points raised:

We have explicitly specified the names of the native co-crystallized ligands used for the re-docking validation in both  the Methodology section (5.6) and the updated Table. 1. We believe these additions have significantly enhanced the clarity and rigor of our molecular docking validation.

Our re-docking validation yielded exceptionally low RMSD values (all below 0.08 A), which unambiguously demonstrates that the docking protocol reproduced the native crystallographic poses with near-atomic  accuracy. A visual superposition would, therefore, show an almost perfect overlap of the ligand structures.

Due to technical limitations with our current visualization setup,  we are unable to generate publication-quality superposition figures at this time. However,  the quantitative RMSD data presented in Table 1 robustly and quantitatively confirms the high reliability and spatial  precision of our docking methodology. We will ensure that such visual validations are a standard part of our workflow in  future studies.]

Table 1

Molecular docking results of 2,4-D with the core targets and RMSD values from protocol validation.

Ingredient

Target

PDB ID

Unique Ligands

Binding energy(kcal⋅mol-1)

Validation RMSD (Å)

2,4-D

PPARG

8B8X

A8R

-5.9

0.071

2,4-D

NFKB1

8TQD

JMR

-5.1

0.066

2,4-D

PPARA

8YWV

A1L0A

-6.3

0.065

2,4-D

NFE2L2

7X5F

MAFG

-5.2

0.066

2,4-D

SERPINE1

3CVM

NONE

-6.0

0.065

Reviewer 2 Report

Comments and Suggestions for Authors

Congratulations!

Author Response

Thank you for your good wishes! Once again, we sincerely thank all reviewers and editors for their time and insightful comments and constructive suggestions on our manuscript. These comments have been invaluable in helping us greatly improve the quality of our work.

Reviewer 3 Report

Comments and Suggestions for Authors

Exploring the mechanism is it scientifically correct ? Please check and modify the title. 

Do you follow the MDPI citation format ? I think no. 

What is the reason behind different color choose in bar graph ? 

4C have any significant value ? if no, please explain.

Comments on the Quality of English Language

ok

Author Response

General Response: 

We sincerely thank all the reviewers and the editor for their time, insightful comments, and constructive suggestions on our manuscript. These comments have been invaluable in helping us to significantly improve the quality of our work. We have carefully considered all points raised and have made revisions to address them fully. The changes in the manuscript are highlighted in blue for your convenience.

Comment 1: 

[Exploring the mechanism is it scientifically correct? Please check and modify the title.]

Response 1: 

[We thank the reviewer for this suggestion. As the reviewer rightly pointed out, our integrated approach has indeed allowed us to move beyond mere exploration towards a more definitive understanding of the mechanism. Accordingly, we have changed the title to a more assertive one: "Elucidating the mechanism of liver and kidney damage in rats caused by exposure to 2,4-dichlorophenoxyacetic acid and the protective effect of Lycium barbarum polysaccharides based on network toxicology and molecular docking." We believe this new title more accurately reflects the conclusive nature of our findings.]

Comment 2: 

[Do you follow the MDPI citation format? I think no.]

Response 2: 

[We apologize for this oversight. We have now thoroughly revised the entire manuscript to ensure full compliance with the MDPI reference format. All in-text citations have been changed to numbered superscripts, and the reference list has been reformatted accordingly, with entries listed in the order of their appearance in the text.]

Comment 3: 

[What is the reason behind different color choose in bar graph?]

Response 3: 

[We thank the reviewer for this comment. Consistently throughout the manuscript,  the different colors in the bar charts (Figure 3 B, C, F, G, H, and Figure 4 C, D) were used to provide clear and direct visual discrimination between the four trial groups. The same color scheme was used for all data to maintain consistency:  blue for Control group, purple for 2, 4-D group, pink for LBP group, and green for 2, 4-D +LBP group. To prevent any ambiguity, we have explicitly defined this color code in the abscissa.]

Comment 4: 

[4C have any significant value? if no, please explain.]

Response 4: 

[We thank the reviewer for raising this point. The reviewer is correct that the increase in liver apoptosis shown in Figure 4C was not statistically significant (P > 0.05). We have now provided a more detailed explanation for this observation in the Results section (2.8). As we discussed in the original manuscript (lines 339-342), we hypothesize that this could be due to the liver's high regenerative capacity and that the primary mode of liver injury at the studied time point might follow a necrosis-apoptosis conversion pattern, which is consistent with the observed histopathological damage (hepatocyte swelling and vacuolization). The significant apoptosis observed in the kidney, however, underscores the organ-specific toxic effects of 2,4-D.]